# Effect of Varying Plasma Powers on High-Temperature Applications of Plasma-Sprayed Al_0.5_CoCrFeNi_2_Ti_0.5_ Coatings

**DOI:** 10.3390/ma15207198

**Published:** 2022-10-15

**Authors:** Sammy Kiplangat Rotich, Ngetich Gilbert Kipkirui, Tzu-Tang Lin, Shih-Hsun Chen

**Affiliations:** 1Department of Mechanical Engineering, National Taiwan University of Science and Technology, Taipei 10607, Taiwan; 2High Entropy Materials Center, National Tsing Hua University, Hsinchu 30013, Taiwan

**Keywords:** HEAs, Al_0.5_CoCrFeNi_2_Ti_0.5_, atmospheric plasma spraying, gas atomization, wear resistance, precipitation strengthening

## Abstract

In this work, the microstructure and mechanical properties of atmospheric plasma-sprayed coatings of Al_0.5_CoCrFeNi_2_Ti_0.5,_ prepared using gas-atomized powders at varying spray powers, are studied in as-sprayed and heat-treated conditions. Gas-atomized powders had spherical shapes and uniform element distributions, with major FCC phases and metastable BCC phases. The metastable BCC phase transformed to ordered and disordered BCC phases when sufficient energy was applied during the plasma-spraying process. During the heat treatment process for 2 hrs, disordered BCCs transformed into ordered BCCs, while the intensity of the FCC peaks increased. Spraying power plays a significant role in the microstructure and mechanical properties of plasma sprayed because at a high power, coatings exhibit better mechanical properties due to their dense microstructures resulting in less defects. As the plasma current was increased from 500 A to 700 A, the coatings’ hardness increased by approximately 21%, which is directly proportional to the decreased wear rate of the coatings at high spraying powers. As the coatings experienced heat treatments, the coatings sprayed with a higher spraying power showed higher hardness and wear resistances. Precipitation strengthening played a significant role in the hardness and wear resistances of the coatings due to the addition of the titanium element.

## 1. Introduction 

In 2004, Yeh [1] and Cantor [2], working independently, proposed high-entropy alloys (HEAs) and multi-principal element alloys (MPEAs), respectively. HEAs typically include five or more elements with equal or near-equimolar quantities and adopt single-phase crystal-lattice structures, such as face-centered cubic (FCC), body-centered cubic (BCC) or close-packed hexagonal (HCP) structures, rather than complex intermetallic compounds [3]. They exhibit unique mechanical properties due to blended effects of severe lattice distortion, sluggish diffusion, high entropy and cocktail effects. Surrounding each atom in the multi-principal-element matrices are distinct kinds of atoms which cause severe lattice distortion. This is considered the main reason for the novel properties of HEAs [3,4,5].

Compared with conventional alloys, several researchers have reported encouraging properties for HEAs, which hold potential in a wide range of applications for structural materials in harsh environments. The AlCoCrFeNi HEA alloy has been widely studied both in bulk and as a surface coating, and has been proposed as a prospective candidate for thermal-barrier and hard-faced coatings [5,6,7]. However, the AlCoCrFeNi alloy is dual phase at room temperature transforms into a single BCC phase at high temperatures, and thus is not suitable for work at high temperatures [8,9]. To investigate the phase stabilities critical for the design of novel HEAs, the Al and Ni contents of the AlCoCrFeNi alloy were adjusted to form Al_0.5_CoCrFeNi_2_ HEA with a stable FCC phase [10,11]. In our previous research, the phase-stable Al_0.5_CoCrFeNi_2_ HEA was alloyed with Ti, forming a major FCC phase and a minor BCC phase [10]. Due to its large atomic radius, the addition of titanium leads to precipitation strengthening, influencing its mechanical properties. The hardness of the BCC precipitates in the Al_0.5_CoCrFeNi_2_Ti_0.5_ gas-atomized powder was 7.85 GPa, while the FCC matrix was 5.3 GPa. Al_0.5_CoCrFeNi_2_Ti_0.5_ is a precipitation-strengthening HEA, and the formation of its phases can be inhibited when prepared in the gas atomization process [10]. 

Additionally, rapidly solidified HEA powders would be frozen at a metastable status, and their properties could be modified by injecting adequate energy. The research on HEAs has focused on bulk materials and to a limited extent on their coatings [11]. The HEAs can be processed as powders using gas atomization or mechanical alloying to avoid the defects encountered in bulk HEAs such as cracks, segregation and shrinkage segregation and some severe defects [9,10,11]. In the mechanical alloying process, impurities and debris from the milling media contaminate the resulting powders, which have poor flowability due to their blocky and angular natures [11]. Alternatively, gas the atomization process is suitable for the production of powders utilized in additive manufacturing processes. In the gas atomization process, powders with a spherical shape controllable size distribution and excellent flowability are ideal for additive manufacturing processes, such as the thermal spray process [7,8]. 

In the thermal spray process, the feedstock material, which could be in form of powders, wires, rods or suspension, is fed into a spray torch and then heated to a molten or near-molten state before being propelled to form a coating on a substrate base material. The classification of thermal spray-coating processes is based on three factors; (1) the combustion heat source, such as high-velocity oxygen fuel spray (HVOF) or a detonation gun; (2) plasma or arc formation using electrical energy, for example an atmospheric plasma spray (APS); (3) low-temperature processes, which utilize energy developed from gas expansion, for instance cold-spray (CS) process [11,12]. The APS process is a versatile process performed under ambient conditions, which makes it economically viable and therefore well established in the surface coatings industry. Additionally, the APS technique is a unique process that offers high deposition efficiency, flexible deposition shapes, onsite repair and cost-effectiveness which has attracted a lot of attention. The critical process parameters in APS that determine the coating quality, density and uniformity include the power input, gas flowrate, carrier gas flow rate, powder feed rate and stand-off distance [7,11]. 

The APS process was chosen to prepare plasma-sprayed Al_0.5_CoCrFeNi_2_Ti_0.5_ coatings in our previous study, where the spraying power and gas flow rate were increased in varying proportions from a current of 500 A and a hydrogen (plasma gas) flow rate of 35 L/min. The study showed that increasing spraying power leads to improved microstructure and mechanical properties of the plasma-sprayed Al_0.5_CoCrFeNi_2_Ti_0.5_ coatings. As a higher spraying power was applied at a current of 750 A and hydrogen flow rate of 50 L/min, a dense microstructure with less defects was observed, which led to an increased hardness from 285 HV_0.3_ at a low spraying power to 396 HV_0.3_. Increasing the spraying power had no effect on the phase structures of the plasma-sprayed coatings as the FCC and minor BCC phases were maintained [10]. 

It is critical to study the impact of high-temperature applications on Al_0.5_CoCrFeNi_2_Ti_0.5_ plasma-sprayed coating’s properties as potential coatings for applications to address harsh engineering environments. This study aims to evaluate the hardness and wear resistances of sprayed Al_0.5_CoCrFeNi_2_Ti_0.5_ coatings as the argon flow rate is kept constant (35 L/min) while the current is varied from 500 A to 700 A when heat-treated at high temperatures. Defects in as-sprayed coatings such as local segregation, pores and oxides could be heat-treated to enhance their hardness and wear-resistance properties at high temperatures.

## 2. Experimental Techniques

### 2.1. Preparation of the Al_0.5_CoCrFeNi_2_Ti_0.5_ Powders

Vacuum induction melting and the inert-gas-atomization process were used for the production of the Al_0.5_CoCrFeNi_2_Ti_0.5_ HEA powders. Based on the designed composition ratio, Al, Co, Cr, Fe, Ni and Ti slugs with a high purity of 99.9% were melted in a high-frequency induction furnace in an argon atmosphere to reduce oxidation. Once the desired melt homogeneity and chemical composition were achieved, the molten material was poured through an insulated tundish and force through a gas nozzle where the molten material was pulverized by high pressure gas in the integrated gas-atomization unit into powders, which were collected by cyclone separation system. The collected powders were categorized into four categories, and the 10–60 µm were chosen for further analysis and production of atmospheric plasma-sprayed coatings [5]. 

The 10–60 µm Al_0.5_CoCrFeNi_2_Ti_0.5_ powders were used to create coatings on a selected AISI 1045 carbon steel substrate measuring 5 × 5 × 0.3 cm^3^ in the atmospheric plasma-spray process using different plasma-spray parameters. The substrates were sandblasted with alumina sand and ultrasonically cleaned in ethanol to remove any form of contamination and dust. Thereafter, the substrates were dried by placing them in the oven for 6 h at 70 °C. 

In atmospheric plasma spray, an electrical current was input and primary and secondary plasma-gas flow rates influenced the formation of the plasma jet temperature and velocity. A mixture of argon as the primary and hydrogen as the secondary gas were utilized and kept at constant flowrates as the current was varied (i.e., 500 A, 600 A, 700 A) to optimize the efficiency of the coating process. Other spraying parameters directly influencing properties of the coating and ultimately the final properties of the coating were the spraying distance, powder feed rate and carrier-gas flow rate. Nitrogen was used as a carrier gas for bringing the alloy powders from the powder feeder to the plasma gun. The spraying parameters of the as-sprayed coatings are as shown in Table 1.

The as-sprayed samples were heat treated in a vacuum furnace at 600, 800 and 1000 °C for a duration of 2 h, respectively, with a heating rate of 10 °C/min. The samples were cooled down to room temperature in the furnace. The as-sprayed coatings were named as P_500_/H_0_, P_600_/H_0_ and P_700_/H_0_, whereas the heat-treated coatings were named based on the spraying power and heat treatment temperature, such as P_700_/_600,_ P_700_/_800_ and P_700_/_1000_, as shown in Table 1

### 2.2. Characterizations

The microstructures of the gas-atomized powders, for both the as-sprayed and heat-treated coatings, were observed using the FE-SEM (JEOL, JSM 7900F, Peabody, MA, USA) equipped with an energy-dispersion spectrum detector (EDS, Oxford Instruments AZtec, High Wycombe, UK). The EDS was used to obtain the chemical compositions of the powders and coatings, while phase structures were analyzed using an X-ray diffractometer (XRD, Bruker D8, Zurich, Switzerland) with Cu Kα radiation (λ = 0.154056 nm). 

The hardness of the as-sprayed and heat-treated coatings were measured using the Vickers microhardness tester (Matsuzawa MMT-X Series, Osaka, Japan) on the cross-sectional area under a load of 0.3 kg, with the dwell time set to 15 s. The hardness values were calculated from the average of eight measurements for each test condition taken on the cross section of each coating sample.

The wear test was performed using a pin-on-disc tribometer (TRB^3^, Anton Paar, Graz, Austria), where the wear resistance of each coating was evaluated using a 6 mm diameter cemented tungsten carbide (WC-6wt% Co) ball counterbody [13]. The load chosen was 3 N applied normally, and the sliding speed selected for the test was 5cm/s on a wear track diameter of 3 mm. The wear length was 100 m for each test. A relative humidity of 50% and an ambient room temperature were the conditions during the wear test. The depth and volume loss of the wear track were estimated using images taken from a white-light interferometry (Filmetrics, San Diego, CA, USA). The wear rate (mm^3^/N·m) was calculated using the following equation.
(1)$$W_{s}=\frac{t}{6b}\left(3t^{2}+4b^{2}\right)2\pi \cdot R$$
(2)$$W_{r}=\frac{W_{s}}{F\times S}$$
where *W_s_* is the wear volume (mm^3^); *t* is the wear depth (mm); *b* is the width of the wear surface (mm); and *R* is the wear track radius (*mm*).

In Equation (2), *W_r_* is the wear rate (mm^3^·N^−1^·m^−1^), *F* is the applied load (N) and *S* is the sliding distance (m). 

## 3. Results and Discussion

### 3.1. Characterization of the As-Sprayed Al_0.5_CoCrFeNi_2_Ti_0.5_ Coatings 

The gas-atomized Al_0.5_CoCrFeNi_2_Ti_0.5_ powders sieved into 10–60 μm in as-obtained condition, as shown in Figure 1a, show identical sizes, highly spherical shapes and wide particle size distribution. The EDS spectrum in Figure 1b shows the distribution of elements in in the EDS spectrum of the selected area as shown in Figure 1a. The particle size distributions of the 10–60 µm powders is shown in Figure 1c, with calculated average diameter as 18 µm. With this shape morphology and particle size distribution as demonstrated in Figure 1, the as-obtained powders are suitable for application in preparation of APS coatings [9,10]. The nominal composition is close to that of the as-atomized powders in Table 2. The Al and Ti elements tend to partition, which could cause the deviation observed between the nominal and as-atomized powders composition [10].

An investigation of the phase constitution of the as-atomized powders shown in Figure 2 revealed major matrix phases of the FCC at, 43.7°, 51.8°, 75.6° and 91.6°, a metastable BCC phase at 43.995°, and disordered BCC at 64.633° and 81.817° [10]. After the application of the powders for the development of the coatings in the APS process wherein the powders reach a molten or near-molten status before being deposited on the substrate, the metastable BCC peak at 43.995 was transformed into an ordered BCC at 35.6°, as shown in Figure 2. Consequently, the disordered BCC peaks can be observed to decrease at 64.633° and completely disappear at 81.817°, which leads to the conclusion that solid-state transformation into ordered BCC at 35.6° occurs. The transformation of the metastable and disordered BCC phase into the ordered BCC phase is facilitated by the high energy of the APS process as the spraying energy increases. 

The macroscopic image of the as-sprayed coating on a AISI 1045 carbon steel substrate is shown in Figure 3. Figure 4a–c shows the SEM images of the as-sprayed coatings at P_500_/H_0_, P_600_/H_0_ and P_700_/H_0_, and it can be observed that the coatings have typical lamellar microstructures with dispersed oxide stringers and some pores. Oxide stringers occur due to the high operating temperature of the APS process (>2000 °C), which causes powder particles to melt and oxidize during the spraying process. Studies have shown that the oxidation of the heated powder particles keeps occurring until the molten particles impact on the substrate, forming lamellar structures with dispersed oxides [12]. The interlamellar cracks, oxide phases and pores decreased as the plasma power was increased, which was attributed to increased spraying power. Researchers have reported reduced levels of porosity as the arc current was increased in the APS process [14]. 

In order to confirm the elemental composition of different regions with typical contrast in the coating, EDS analysis was performed, as shown in Figure 4c. The results show that the gray phase in Figure 4a–c points one, three and five were rich Co, Cr and Ni elements but not for Al element, while the dark gray dispersed points two, four and six were of Ti and Cr mixed oxides. Based on the presented XRD analysis, regions one, three and five correspond to the FCC matrix, disordered and ordered BCC, respectively [14]. Although the point EDS analysis showed the presence of oxidized titanium, XRD was not able to detect their diffraction peaks due to their low contents [14,15]. 

The hardness of the coating increased as the spraying power was increased from P_500_/H_0_ to P_700/_H_0_, as shown in Figure 5. The average thickness of the coatings after grinding and polishing varied between 95 µm to 110 µm, and the measurements were taken using the SEM images, as shown in Figure 5b. The as-sprayed coatings could have been in a supersaturated state because of the high spraying temperature and the rapid cooling rate, and therefore the properties of the powder particles could have transferred to the coatings during the APS process. The powder particles’ qualities and morphologies influence properties of the coatings, such as density and, ultimately, integrity [9]. Studies have shown that gas-atomized powders produce dense coatings which can be attributed to the spherical particles and uniform size distributions [6]. The precipitation-strengthening phenomenon due to dispersed Ti_2_O which form oxide stringers, as shown in Figure 4, impact on the hardness of the coatings. A denser coating is formed by higher plasma-spraying power, as it enables a near-complete melting of the powders which has less defects, resulting in a higher hardness. Less of the oxide stringers could be observed as the spraying power was increased, as shown in Figure 4a–c, possibly increasing the hardness. These coatings can withstand high-temperature applications, as demonstrated by other researchers [14]. 

### 3.2. Characterization of Heat Treated Plasma Sprayed Al_0.5_CoCrFeNi_2_ Coatings

Figure 6 shows the XRD spectrum of the coatings at different temperatures which kept the FCC phase and minor-ordered BCC. The disordered BCC phase (64.633°) in the as-sprayed coatings discussed in Figure 2 precipitated into ordered an BCC peak at 35.6 due to the heat treatment process and longer cooling period. The precipitation of the ordered B2 structure from disordered BCC could be attributed to different bonding enthalpies for Al-Ti (−30 kJ/mol), Al-Ni (−22 kJ/mol) and Al-Co (−19 kJ/mol) compared with Al-Cr (−10 kJ/mol) and Al-Fe (−11 kJ/mol). Research on AlCoCrFeNiTi_0.5_ showed that ordered B2 is Al-, Ti- Ni-rich and depleted in Fe and Cr [16]. Both the gas-atomized powders and the as-sprayed powders were rapidly cooled, and therefore remained in supersaturated states. The phase stability of this alloy at high temperatures could be attributed to high configurational energy [9]. The diffraction peaks narrowed with increasing temperature, and the intensities of the coatings heat-treated at above 800 °C were enhanced. As the heat treatment temperature increased, the average grain sizes increased from 9.42 µm at 600 °C to 16.5 µm at 800 °C and 20.54 µm at 1000 °C when calculated using the Scherrer equation. This could be attributed to grain growth as the temperature kept rising. The intensity of the XRD peaks growths narrowed and rose as the heat treatment temperature was increased due to improved crystallinity. Similar observations were reported for CoCrFeNiMn HEA coatings annealed at 900 °C [17]. The increasing grain size is evidence that grain growth occurs after heat treatment at high temperatures. 

The overview of the microstructure in Figure 7a–f is homogenous with some black oxide stringers dispersed in the coating after the heat-treatment process for 2 h at different temperatures. The stacking deposition of molten metal droplets generated a typical lamellar structure of HEA coating, as shown in Figure 4 for the as-sprayed coatings, which was displayed more clearly after heat treatment at different temperatures. However, notable changes occurred in the coating’s microstructure at different temperatures, such as interlamellar cracks filled by Ti and Al oxides as the heat-treatment temperature increased. The chemical compositions of each coating are summarized in Table 3, where the light gray areas (a–k) are rich in Ni, Co and Fe while the dark gray areas (b–l) are a mixture of Ti and Al oxides. Gray areas forming the matrix (a–q) could be the major FCC phase (Fe-Cr) and minor-ordered BCC phase (Al-Ni-Ti), while the dark gray areas (b–l) could be the titanium-rich oxide based on the XRD analysis in Figure 2 and Figure 6 [14,16]. The oxides were not detected in the XRD spectrum due to their low contents and due to the fact that the heat-treatment process was conducted under vacuum pressure. 

The microstructure at 600 °C for different plasma powers in Figure 7a,d, showed no change in the size of oxides and that the interlamellar cracks widened after heat treatment in comparison with the as-sprayed coatings in Figure 4a,b. A higher magnification of the coatings sprayed at a high spraying power heat treated at 600 °C results in interlamellar cracks, especially between the matrix and the oxide particles, as shown in Figure 7a. As the heat-treatment temperature increased to 800 °C, the lamellar structures of the coatings were observed to be dense, as depicted in Figure 7b. Titanium, which diffuses quickly to form oxides that fill the interlamellar cracks and pores, could explain the change in the coatings’ morphologies as the heat treatment was increased from 600 °C to 800 °C and 1000 °C. Additionally, titanium promotes the precipitation-strengthening phenomenon. Both Ti and Al are BCC stabilizers, and thus the addition of Ti to Al_0.5_CoCrFeNi_2,_ which was purely FCC, brought about the precipitation of a BCC phase in Al_0.5_CoCrFeNi_2_Ti_0.5_ [8,14,16]. The microstructural changes due to different heat-treatment temperatures influenced the mechanical properties of both the as-sprayed and heat-treated coatings [17]. 

Other than oxides, cracks and interlamellar gaps, pores are observed in Figure 4 and Figure 7. Generally, porosity levels in atmospheric plasma-sprayed coatings vary between 2% to 20% depending on the spraying parameters and the characteristics of the powders. Studies have shown that by increasing the hydrogen flowrate from 7 L/min to 9 L/min, a coatings’ hardness can increase by 150 HV_0.5_ while porosity decreases by 3%. The decrease in porosity is due to the improved melting state of the powders [18]. Decreasing the hydrogen flow rate increases the porosity levels of the coatings, according to studies by Zhang et al., [19]. The electrical input has a similar effect as the hydrogen gas flow rate on the porosity and mechanical properties of the coatings. By optimizing the spraying parameters, lower porosity levels of 4.4% were achieved using response surface methodology [20].

## 4. Mechanical Properties 

### 4.1. Hardness of the Heat Treated Coatings

The as-sprayed coatings presented in Figure 4 show that an increase in plasma power leads to an increase in hardness, which has been re-plotted in Figure 7 to demonstrate the comparison. Heat treatments performed across the three temperature states show increased hardness particularly at 600 °C and 800 °C, as demonstrated in Figure 8. Heat-treated coatings kept a similar trend, whereas coatings sprayed with a higher spraying power had a higher hardness, up to 517.3 HV_0.3_, 551.6 HV_0.3_, and 429.3 HV_0.3_ at 600 °C, 800 °C and 1000 °C, respectively, which is attributed to the dense stacking and structure properties of the powders being transferred to the coatings, as demonstrated in Figure 4. A higher plasma spraying power raises the working temperature and ultimately leads to a dense and continuous coating. Additionally, precipitation strengthening plays a critical role in the increased hardness of the as-sprayed and heat-treated coatings, as the properties inherent in the gas-atomized powders are transferred to the coatings [8]. The enhanced hardness properties of the coatings performed at high plasma powers are due to better bonding leading to less defects. However, the presence of voids and interlamellar cracks observed in Figure 7a,d for the heat-treated coatings at 600 °C impacts on the hardness of the coatings [9,21]. The presence of pores and poor adhesion between oxides and the HEA matrix may reduce the cohesive strength, which negatively impacts on hardness [21]. The Ti and Al elements diffuse into interlamellar cracks forming Ti-oxides after annealing at 800 °C, according to the EDS spot analysis for the point in Figure 7c. 

The intense diffusion of Ti-rich oxides to the interlamellar gaps increased the hardness of the coatings annealed at 800 °C. However, as the temperature was increased to 1000 °C there was a decrease in hardness, which can be attributed to crystallization and grain growth. The sharp narrow peaks of the coatings annealed at 1000 °C presented in Figure 6 show that annealing treatments promote crystallization and grain growth, which lowers the hardness of the coatings. The intensities and widths of the XRD peaks in Figure 6 increase and decrease, respectively, as the heat-treatment temperature is increased from 600 °C to 1000 °C. Based on the XRD spectrum in Figure 6 and using the Scherrer equation, it can be calculated that the approximate grain size increased from 9.42 at 600 °C to 16.5 µm at 800 °C and 20.54 µm at 1000 °C. Precipitation hardening at high temperatures also occurred in the alloy coatings, and the 1000 °C heat-treated coatings showed lower hardness than the coatings at 600 °C. The overall increase in microhardness may be attributed to the increase of the cohesive strength among the splats and oxides contents, as observed by Xiao et al. [9].

### 4.2. Wear Resistance 

The wear rates of the as-sprayed and heat-treated coatings are showed in Figure 9, where spraying powers with varying temperature conditions are compared. The as-sprayed coatings had the highest wear rate as compared with the heat treated coatings, which is consistent with the hardness data reported in Figure 8, whereas the coatings heat-treated at 600 °C and 800 °C had the lowest wear rates of below 1.8 × 10^−5^ mm^3^N^−1^m^−1^. Consequently, the coatings heat-treated at 1000 °C showed higher wear rates of 2.25 × 10^−5^ mm^3^N^−1^m^−1^, 2.88 × 10^−5^ mm^3^N^−1^m^−1^ and 2.7 × 10^−5^ mm^3^N^−1^m^−1^. The wear rates of the coatings were directly proportional to the hardness of the coatings, and therefore coatings that showed higher hardness were observed to demonstrate low wear rates. The precipitation-strengthening mechanism in Al_0.5_CoCrFeNi_2_Ti_0.5_ attributed to the presence of titanium enhanced the hardness of the powders and subsequently the coatings [8,14]. The low wear resistances of the as-sprayed coatings could be attributed to their supersaturated states due to rapid cooling, and therefore the precipitation strengthening was not complete. As heat treatment was performed on the coatings, the precipitation strengthening process occurred which increased the hardness of the coatings, particularly at 600 °C and 800 °C. However, the grain growth at 1000 °C from the XRD data in Figure 6 decreased the hardness of the coating and ultimately the wear-resistance properties [9]. Figure 10 shows the profilometer wear tracks of the images of the heat-treated coatings for higher spraying powers. The observations from Figure 10 are consistent with the calculated wear rates, which demonstrates that the coatings heat-treated at 1000 °C had low wear resistances. 

The Al_0.5_CoCrFeNi_2_Ti_0.5_-sprayed coatings could be suitable for applications at high temperatures, particularly at 600 °C to 800 °C as they possess higher hardness which are directly proportional to the wear-resistances of the materials [14,22]. Although most researchers have showed that better properties are obtained by heat treatments at 600 °C, in this study, the coatings heat-treated at 600 °C as shown in Figure 7a,d had defects such as interlamellar cracks, which impacted negatively on their hardness and wear resistances [23]. The coatings heat-treated at 800 °C showed higher hardness and wear resistances as they had homogenous microstructures with minimal defects, which points to improved cohesive strength between the matrix, precipitates and oxides, as shown in Figure 7b,e.

The worn surfaces of the heat-treated coatings at low and high spraying powers are shown in Figure 11, along with varying features. The wear tracks in Figure 11a,c,d,f are rough with pits on the surface and delaminations, while (b) and (e) are relatively smooth with shallow spalling of the tribolayer. Figure 11a,d for the coatings heat-treated at 600 °C show tribolayer in some parts, but also large delaminations which could be attributed to the interlamellar cracks observed in Figure 7a,d. The cohesive strengths for these coatings could be affected by the interlamellar cracks, and this could be the main reason for the wear. The tribolayer formed by repeated gliding of the hard WCO ball eventually fails by forming splats [9]. The coatings heat-treated at 800 °C showed large areas covered by tribolayers which increased their wear resistances, as shown in Figure 11b,e. The EDS analysis of the tribolayer in Figure 11a depicts lower amounts of oxides (Al-2.2, Ti-8.2, Cr-8.7, O-10.0, Fe-12.2, Co-17.6, Ni, 41.1) wt%. Some cracks are observed which are perpendicular to the sliding direction, which could be fatigue cracks from the repeated rubbing action between the hard WCO ball and the coating, which eventually leads to delamination. In Figure 11c,f, the adhesion between the counterbody and the coating could have occurred which led to adhesive wear [24]. Additionally, the serrated ridges perpendicular to the direction of sliding suggest severe adhesion wear. Shallow grooves can be observed in Figure 11c,f which suggest abrasion wear. Both the adhesive and abrasive wear mechanisms played a role in the degradations of the coatings. 

## 5. Conclusions

In this study, atmospheric plasma spraying was utilized to produce Al_0.5_CoCrFeNi_2_ coatings which were analyzed in as-sprayed and heat-treated conditions. The following conclusions can be drawn from this study; 

As the spraying power increases, a dense coating is formed with increased hardness and wear resistance, and the same trend is kept in heat treated coatings. The average hardness of the as-sprayed coating for P_500_/H_0_, P_600_/H_0_ and P_700_/H_0_ were 307 HV_0.3_, 335 HV_0.3_ and 372 HV_0.3_, respectively, which increased after heat treatment at higher temperatures, particularly at 600 °C and 800 °C by an average of 25%, which could be attributed to precipitation strengthening.The precipitation-strengthening phenomenon due to the addition of Ti increased the mechanical properties of the coatings at high temperatures. The coatings heat-treated at 600 °C and 800 °C showed high resistances, which was consistent with their hardness trends.The wear mechanisms on the heat-treated coatings were both adhesive and allowed for some minor observations of abrasive wear.

## Figures and Tables

**Figure 1 materials-15-07198-f001:**
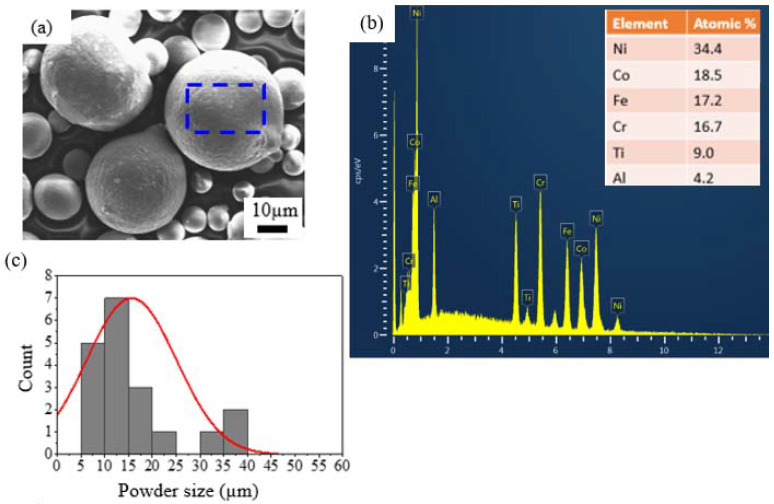
The 10–60 µm gas-atomized Al_0.5_CoCrFeNi_2_Ti_0.5_ powders (**a**) SEM image morphology and distribution, (**b**) EDS spectrum of the selected area shown in (**a**), (**c**) normal distribution graph for the particle size.

**Figure 2 materials-15-07198-f002:**
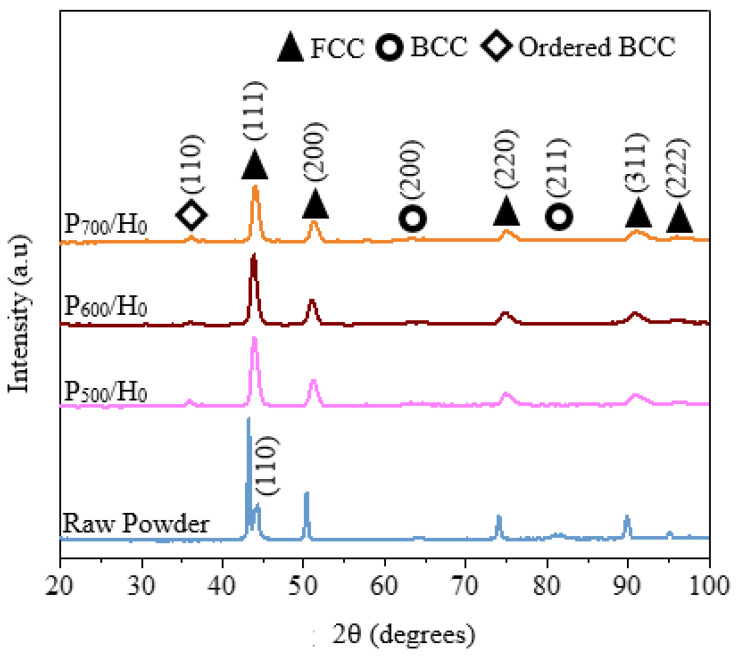
XRD spectrum for of the 10–60 µm powder and as-sprayed coatings at varying spraying powers from P_500_/H_0_ to P_700_/H_0_.

**Figure 3 materials-15-07198-f003:**
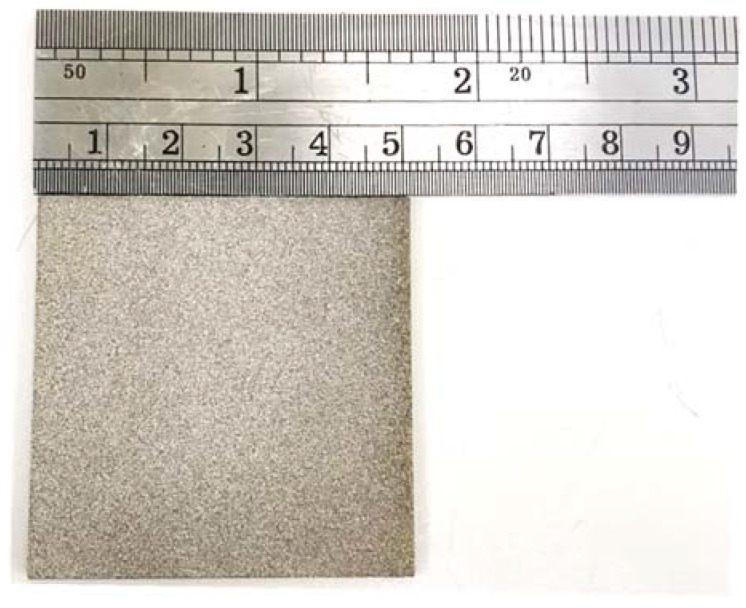
Macroscopic image of the Al_0.5_CoCrFeNi_2_Ti_0.5_-sprayed coatings.

**Figure 4 materials-15-07198-f004:**
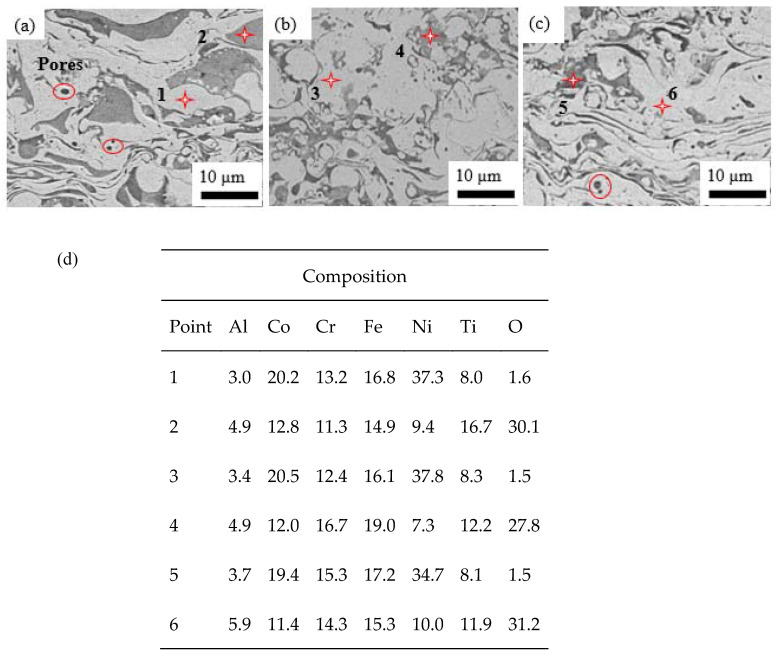
SEM microstructure of the as-sprayed coatings: (**a**) low power, P_500_/H_0_; (**b**) medium power P_600_/H_0_ (**c**) high power, P_700_/H_0_ (**d**) Elemental composition at P_500_/H_0_ P_600_/H_0_, and P_700_/H_0_.

**Figure 5 materials-15-07198-f005:**
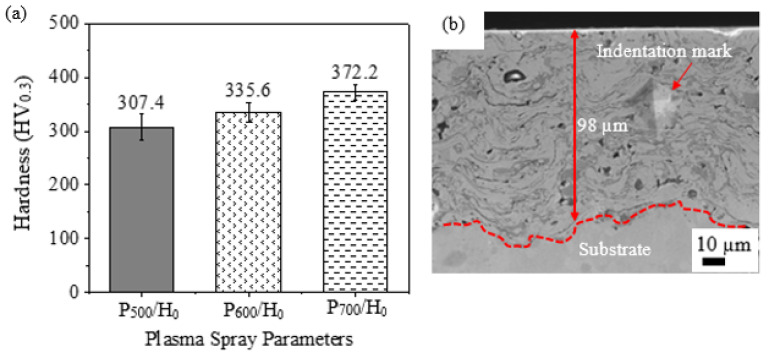
(**a**) Hardness of the as-sprayed coatings with varying spraying powers and (**b**) SEM cross-sectional image of the P_500_/H_0_ coating.

**Figure 6 materials-15-07198-f006:**
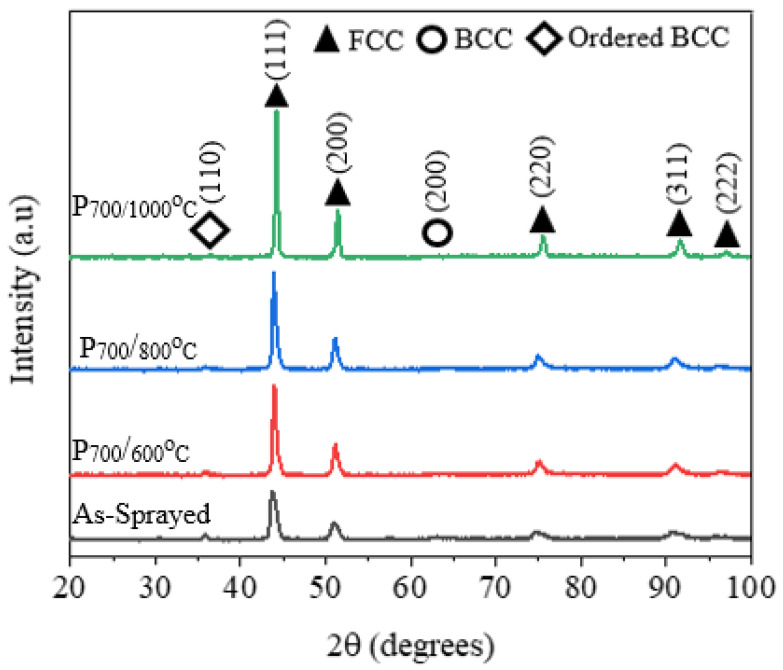
XRD spectrum of the heat-treated coatings with a spraying power of P_700_.

**Figure 7 materials-15-07198-f007:**
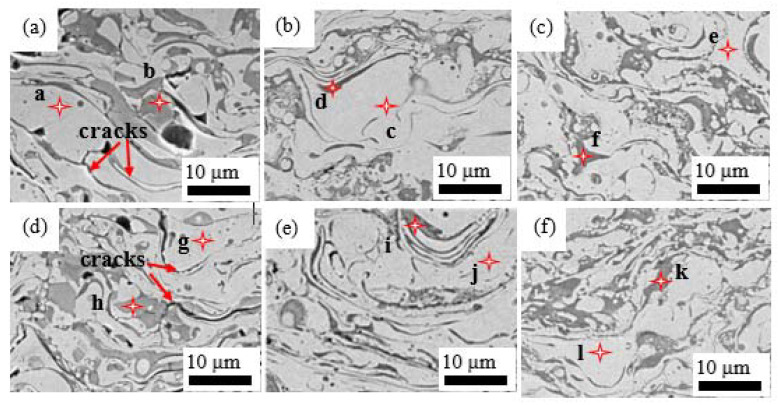
Cross-sectional images for different plasma-power heat-treated at different temperatures; (**a**) P_500_/_600 °C_, (**b**) P_500_/_800 °C_, (**c**) P_500/1000 °C_, (**d**) P_700/600 °C_, (**e**) P_700/800 °C_ and (**f**) P_700/1000 °C_.

**Figure 8 materials-15-07198-f008:**
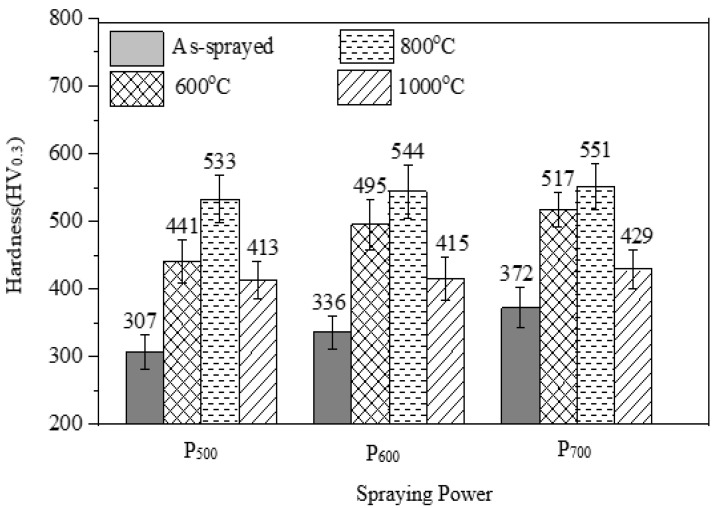
Average hardness of the as-sprayed and heat-treated coatings at 600 °C, 800 °C and 1000 °C.

**Figure 9 materials-15-07198-f009:**
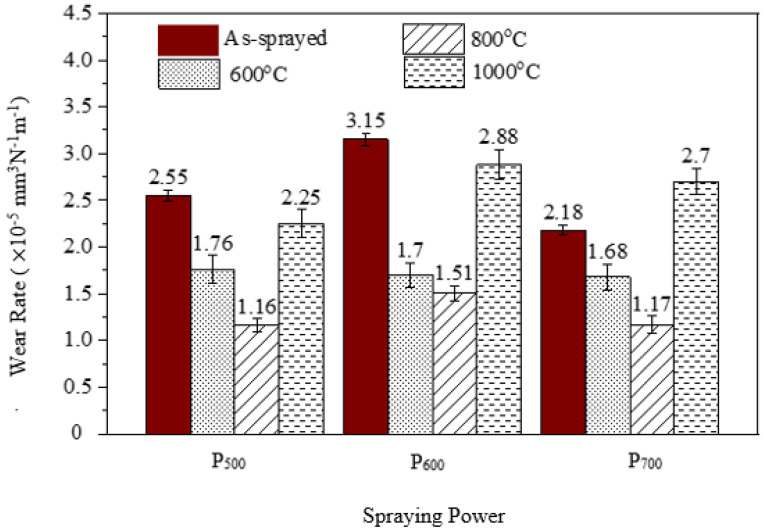
Volumetric wear rate of the as-sprayed and heat treated Al_0.5_CoCrFeNi_2_Ti_0.5_ HEA.

**Figure 10 materials-15-07198-f010:**
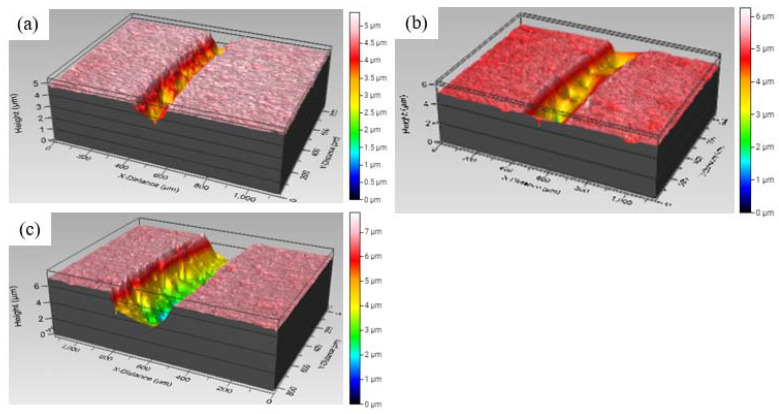
Three-dimensional profilometer wear track images for (**a**) P_700/600 °C_, (**b**) P_700/800 °C_ and (**c**) P_700/1000 °C_.

**Figure 11 materials-15-07198-f011:**
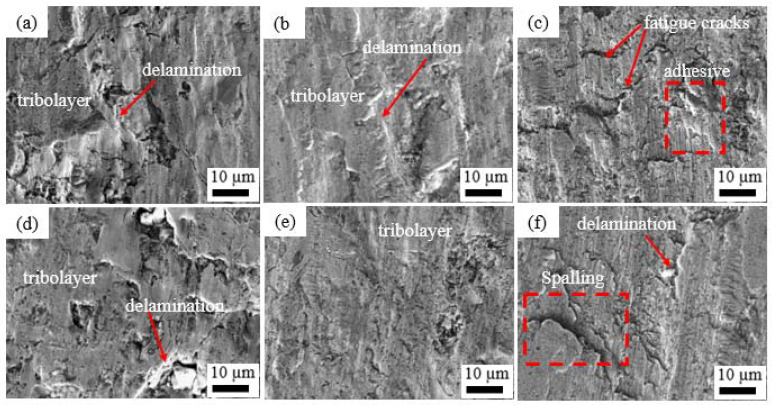
SEM images of the worn surfaces for the heat-treated coatings: (**a**) P_500/600 °C_; (**b**) P_500/800 °C_; (**c**) P_500/1000 °C_; (**d**) P_700/600 °C_; (**e**) P_700/800 °C_ and (**f**) P_700/1000 °C_.

**Table 1 materials-15-07198-t001:** Parameters of the plasma-spraying process for the Al_0.5_CoCrFeNi_2_Ti_0.5_ coatings.

Sample	Plasma Setup				Spray Distance (mm)	Traversing Speed (mm/s)	Feed Rate (mm)
	Current (A)	Ar (L/min)	Voltage (V)	H_2_ (L/min)			
P_500_	500	35	70.0	9.3	140	750	30
P_600_	600						
P_700_	700						

**Table 2 materials-15-07198-t002:** The compositions of the nominal elements and as-atomized Al_0.5_CoCrFeNi_2_Ti_0.5_ powder.

Element	Al	Co	Cr	Fe	Ni	Ti
Nominal	8.3	16.7	16.7	16.7	33.3	8.3
10–60µm	4.2	18.5	16.7	17.2	34.4	9.0

**Table 3 materials-15-07198-t003:** EDS results of the annealed coatings at different temperatures as marked area in Figure 5.

		Composition (at%)						
Coatings	Point	Al	Co	Cr	Fe	Ni	Ti	O
P_500/600 °C_	a	3.4	20.2	15.1	17.3	35.7	7.4	1.0
	b	5.4	13.7	12.4	13.6	13.0	12.5	29.4
P_500/800 °C_	c	2.4	20.6	10.1	14.7	44.7	6.0	1.4
	d	19.0	0.7	2.6	1.1	1.4	37.0	38.1
P_500/1000 °C_	e	1.5	21.2	12.9	18.5	39.9	4.3	1.8
	f	15.7	1.0	5.4	1.7	1.1	35.2	39.8
P_700/600 °C_	g	2.9	20.8	13.1	16.7	37.9	7.7	1.0
	h	6.6	6.3	18.6	15.6	5.2	18.0	29.6
P_700/800 °C_	i	9.9	3.8	5.4	4.2	6.4	39.8	30.5
	j	1.9	20.4	10.1	16.0	44.4	6.6	1.1
P_700/1000 °C_	k	0.6	23.4	12.5	18.2	41.8	2.4	1.1
	l	7.1	4.3	21.3	14.2	3.7	21.7	27.7

## Data Availability

The data presented in this study are available on request from the corresponding author.

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
