# Peer review of "Effect of Varying Plasma Powers on High-Temperature Applications of Plasma-Sprayed Al0.5CoCrFeNi2Ti0.5 Coatings"

_materials, 2022, doi:10.3390/ma15207198_

Round 1

Reviewer 1 Report

Typo errors should be re-checked and corrected.

Reviewer 2 Report

The authors provided an interesting study about the effect of plasma power on the coating of high entropy alloys. However, there are some methodological and technical aspects that should be clarified. The comments can be found in the annotated manuscript.

Reviewer 3 Report

In terms of content, the work interesting, but in terms of editing (text, graphics, units, references) there are a lot of shortcomings in fact chaotically prepared. In general, the substantive content of the article I evaluate positively, but during the reading of the article I have come up with some questions and comments. Please respond to the ones listed below:

1. In the Abstract section, the authors use the incorrect phrase "films' hardness." This is not correct, as "films" refers to much smaller thicknesses than those addressed in this paper.

2. The way of citation as: line 53 [10-15] (6 cited items in one sentence) and line 76 [9-18] (10 cited items in one sentence) is not acceptable.  Many of the above literature items e.g. 11, 13, 14, 15 are no further referenced in the text. Please that there is a precise reference to each of the cited works in the text, alternatively remove them from the references items.

3. Carelessly and inconsistently stated units of measurement and especially temperature among others see line 106, 229, 273 etc. It makes no difference to authors whether there is a space after a numerical value or not see e.g. sliding speed, diameter gas-atomized, current ect. Please carefully review the entire text and correct according to the art of specifying units.

4. In Table 1 it is written H2 but it should be with subscript H2

5. Please indicate what changes in average grain sizes were obtained for samples heat treated at 800°C. (see lines 227÷229).

6. By analyzing Fig. 3 and Fig. 6, the question arises what is the porosity of the tested coatings. Please include such information in the text of the manuscript.

7. The authors arbitrarily and interchangeably use the terms microhardness and hardness (see title Fig. 4 and Fig. 7, lines 313 and 362). According to the standard, Vickers hardness testing at HV0.3 is still hardness, not microhardness.

8. In the test, the description regarding the worn Surface analysis (line 346) erroneously refers to Figure 8. In Fig. 9 show photographs of the worn surface for other tested sample variants. The authors limited themselves to only P700/800°C and P700/1000°C samples. In addition, please identify and compare what wear mechanisms occur on the surface during wear sliding.

Round 2

Reviewer 3 Report

Bad wear factor unit. There should be mm3 N-1 m-1. See  title Fig. 9 and text in Wear Resistance section.

According to the standard, Vickers hardness testing at HV0.3 is still hardness, not microhardness (see title Fig. 5).
